# LSD-NET: LOOK, STEP AND DETECT FOR JOINT NAVIGATION AND MULTI-VIEW RECOGNITION WITH DEEP REINFORCEMENT LEARNING

## ABSTRACT

Multi-view recognition is the task of classifying an object from multi-view image sequences. Instead of using a single-view for classification, humans generally navigate around a target object to learn its multi-view representation. Motivated by this human behavior, the next best view can be learned by combining object recognition with navigation in complex environments. Since deep reinforcement learning has proven successful in navigation tasks, we propose a novel multi-task reinforcement learning framework for joint multi-view recognition and navigation. Our method uses a hierarchical action space for multi-task reinforcement learning. The framework was evaluated with an environment created from the ModelNet40 dataset. Our results show improvements on object recognition and demonstrate human-like behavior on navigation.

## 1 INTRODUCTION

Object recognition is an important task for robot navigation, object manipulation and scene understanding. While traditional methods use single-view images for object recognition, some recent methods have proposed to use multi-view image sequences. We believe that the multi-view representation is more realistic than a single-view representation for object recognition. Also, we further believe that all viewpoints are not necessary for this task because few specific viewpoints can effectively classify the object. Therefore, the task is to find the most effective multi-view image sequence for recognition, which comes under the umbrella of active vision (Ammirato et al., 2017) i.e. use the current input to further improve the understanding.

Active multi-view recognition uses current information to make an approximate guess of the next best view to better understand an object. This mimics human behavior for various recognition tasks. When an object is occluded or unrecognizable from a viewpoint, humans hypothesize and move to next best view for improved object recognition. Similarly, humans can rotate or move objects for a multi-view image sequence, which results in better learning and recognition. Hence, navigation can be naturally introduced into multi-view object recognition. We combine the tasks of object classification from multiple views with navigation into a joint framework.

Navigation has been a challenging problem tackled by different learning frameworks. Recent works show that navigation is deeply ingrained in the deep reinforcement learning framework (Zhu et al., 2017). We combine the ideas of navigation, object recognition, and deep reinforcement learning in a coherent manner to teach an agent how to navigate a scene and classify a target instance simultaneously. With the recent success in navigation, deep reinforcement learning is a natural solution to this classification problem augmented with navigation. Ideally, instead of an inductive bias like image pairs (Johns et al., 2016), the agent would learn how to navigate the scene, like which direction to move or the number of steps to take, and learn an object's multi-view representation that results in greatest probability of correct classification. Navigation and exploration are key traits to learn the generalized multi-view representations, and deep reinforcement learning helps in performing them.

Figure 1 visualizes the overall framework. In the beginning, the agent is uncertain of the object. Consequentially, it moves in the direction that would maximize its certainty. As the agent maneuvers the scene, it learns the multi-view representation of the object and creates hypotheses of the its

category. The agent classifies once it becomes certain after a number of steps. In the best case scenario, the agent can classify the object given the first image it sees, although exploration and movement are highly desirable traits in cases of uncertainty.

Another important aspect of the framework is the memorization of previous object views. Similar to a human's memory of previous information, data from previous images are combined to learn better actions and improve the prediction accuracy. The other advantage of our method is independent of the starting point. We can start at any random point of the environment and navigate to accurately classify the object. We propose the first framework to combine learning of navigation and object recognition, which are widely considered as two different tasks, as a form of multi-task learning.

## 2 RELATED WORK

**Object recognition:** In the domain of computer vision and machine learning, multiple methods have been proposed for multi-view object recognition. Most of the recent papers in this direction use convolutional neural networks (CNN) for this task. 3D ShapeNets (Wu et al., 2015) proposes to use 3D features as input to a 3D CNN and showed state-of-the-art object recognition using 3D objects. They further showed next best view selection for 3D object recognition as an auxlary task of their pipeline. Multi-view CNN (Su et al., 2015)(Qi et al., 2016) have generalized multi-view recognition by learning from images that cover the full sphere of viewpoints over an object. Multiple methods have exploited the of multi-view recognition in 3D and 2D to improve the accuracy (Qiu & Yuille, 2016). Some recent works also proposed a reinforcement learning based method to solve the problem of active vision by looking ahead (Jayaraman & Grauman, 2016) and have shown its applications to object recognition. Further a pairwise decomposition method has been proposed to learn the best pairs to recognize the object (Johns et al., 2016). Although most of the methods have addressed the problem with different approaches, they introduce prior knowledge and bias in how to search for the next best view to recognize an object. We believe navigation with reinforcement learning as the best method to learn the next best view selection. There is work proposed in object detection with visual attention (Mnih et al., 2014), which uses reinforcement learning over glimpses of a image to predict the label.

**Navigation:** Navigation with reinforcement learning is an extensively addressed problem in robotics domain. The deep Siamese actor-critic networks (Zhu et al., 2017) handles the problem of multiple targets and scenes generalization for navigation. Also, recent literature show promising results for visual navigation by using cognition mapping and planning (Gupta et al., 2017). They use value iteration networks (Tamar et al., 2016) to learn the policy for navigation. In addition, there has been motivation of navigation for object instance recognition in real-world environments (Ammirato et al., 2017). With the introduction of deep learning to reinforcement learning, there has been a plethora of recent advancements in understanding the how humans navigate and interact with the environment. Recent methods in deep reinforcement learning like the Deep Q-Network algorithm (Mnih et al., 2013) and the Asynchronous Advantage Actor Critic (Mnih et al., 2016) algorithm showed near human level performance on multitude of tasks like Atari 2600 games (Mnih et al., 2013). Furthermore, deep reinforcement learning showed state-of-the-art results for playing games like Atari (Mnih et al., 2013), Go (Silver et al., 2016), and Doom (Mirowski et al., 2016). With the advent of recent successful reinforcement learning algorithms in navigation, research has shifted towards real world applications of navigation and scene understanding with deep reinforcement learning.

## 3 METHOD

The methods proposed above learns navigation and recognition as unrelated individual tasks. Although these methods show promising results in the direction of active vision, the task of end-to-end navigation and recognition has not been explored. Hence, we propose the current model to learn object categorization from navigation.

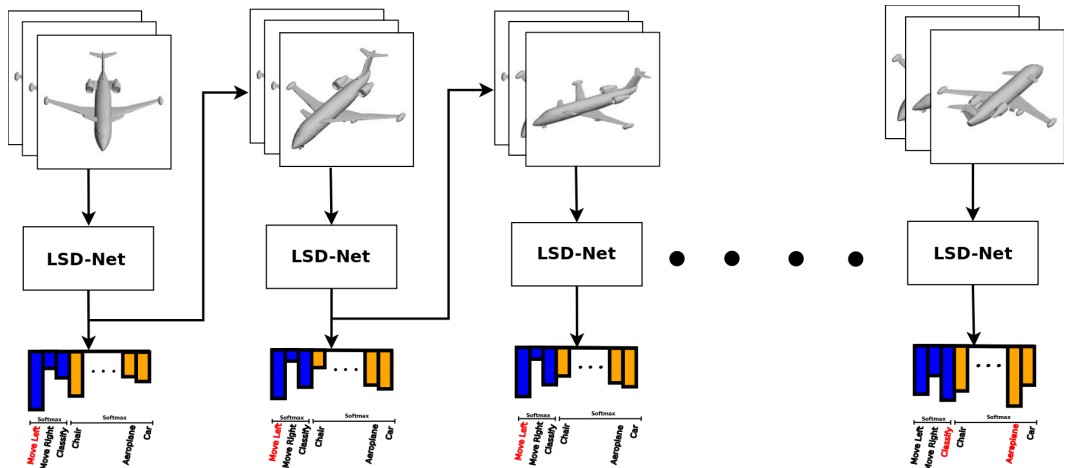

Figure 1: The illustration of the current pipeline for object centric navigation. The stacked images are passed to the network and the network has to decide whether to navigate or classify the object. Once the classification is achieved the environment stops.

## 3.1 BACKGROUND

In a standard reinforcement learning setting, an agent interacts with an environment over a number of discrete time steps. At each time step $t$, the agent observes a state $s_t \in \mathcal{S}$ and selects an action $a_t \in \mathcal{A}$ where $\mathcal{S}$ and $\mathcal{A}$ are the state and action spaces, respectively. Guided by a policy $\pi$ that maps states $s_t$ to actions $a_t$, the agent performs an action $a_t$ and observes the next state $s_{t+1}$ and reward $r_t$.

To compute the policy $\pi$, neural network function approximators can be used to calculate $\pi(a_t|s_t; \theta)$ for some function parameters $\theta$. The REINFORCE algorithms (Williams, 1992) are policy-based model-free methods that can update $\theta$ by gradient ascent on $E[R_t]$, where $R_t = \sum_{k=0}^{\infty} \gamma^k r_{t+k+1}$ is the accumulated discounted reward and $\gamma \in (0, 1]$ is the discount factor.

Also with state $s_t$, a baseline value $b_t(s_t)$ can also be computed with neural network function approximators for some function parameters $\theta$. One baseline can be the state value function estimate $v(s_t; \theta)$. In the actor-critic learning architecture (Sutton & Barto, 1998), the policy $\pi$ and the baseline $b$ are the actor and critic, respectively. Then the overall loss function is:

$$l(\theta) = l_\pi(\theta) + l_v(\theta) \tag{1}$$

which is the sum of the policy loss $l_\pi(\theta)$ and value loss $l_v(\theta)$. The individual policy and value loss functions, as shown by Equation 2 and Equation 3 respectively, are:

$$l_\pi(\theta) = log(\pi(a_t|s_t; \theta))(R_t - b_t(s_t)) \tag{2}$$

$$l_v(\theta) = \frac{1}{2}(R_t - b_t(s_t))^2 \tag{3}$$

With learning rate $\eta$, differentiation of the loss function in Equation 1 with respect to the weights $\theta$ results in a gradient descent update $\theta_{i+1} \leftarrow \theta_i - \eta \nabla_{\theta_i} l(\theta_i)$, which is shown to update the function approximate $\pi(a_t|s_t; \theta)$ towards the optimal policy $\pi^*(a_t|s_t)$ (Williams, 1992).

## 3.2 LSD-NET: LOOK, STEP AND DETECT NETWORK

We propose a novel architecture to simultaneously learn to navigate and recognize an object. The main contribution of this architecture is the hierarchical action space and extrapolation of the action space to a higher dimension for multi-task learning.

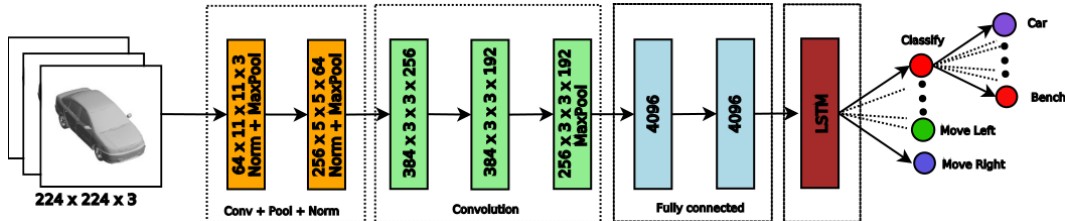

Figure 2: The proposed architecture for learning the object centric navigation. We used AlexNet and combined it with our novel hierarchical actions to simultaneously navigate and recognize the object.

Recent multi-view recognition methods use deep neural networks, like pairwise decomposition (Johns et al., 2016) used a VGG-M (Chatfield et al., 2014) network. Motivated from their success in multi-view recognition, we use a network architecture similar to AlexNet (Krizhevsky et al., 2012) which contrasts to the simple deep neural networks used by most of the deep reinforcement learning methods. This need for deeper networks is rooted from the recent success of image classification tasks using these networks.

Similar to AlexNet, our architecture's first convolutional layer filters the 224 x 224 x 3 input of stacked frames with 64 kernels of size 11 x 11 x 3 with a stride of 4 pixels. The second convolutional layer takes the normalized and max-pooled output of the first convolutional layer as input and filters it with 256 kernels of size 5 x 5 x 64. The third convolutional layer takes the normalized and max-pooled outputs of the second convolutional layer as input and filters it with 384 kernels of size 3 x 3 x 256. The fourth convolutional layer takes the outputs of the third convolutional layer as input and filters it with 384 kernels of size 3 x 3 x 192. For the last convolutional layer, the fifth convolutional layer takes the outputs of the fourth convolutional layer as input and filters it with 256 kernels of size 3 x 3 x 192. The first and second fully-connected layers have 4096 units each, where the first fully-connected layer takes the outputs of the fifth convolutional layer as input and the second fully-connected layer takes the outputs of the first fully-connected layer as input.

### 3.3 HIERARCHICAL ACTION SPACE AND POLICY

A single softmax layer is sufficient for simple action spaces, like movement actions of some Atari 2600 games (Mnih et al., 2013). Though in complex action spaces such as classification and navigation tasks, a single softmax layer is biased towards tasks with larger number of actions. We propose a hierarchical action tree based method to reduce the bias in the action space.

If there are $m$ object categories and $n$ movements, then let $\mathcal{C} = \{a_1, ..., a_m\}$ be the actions that classifies the object as one of the $m$ categories and let $\mathcal{N} = \{a_{m+1}, ..., a_{m+n}\}$ be the actions that navigate the scene. Then let the actions $a_{\mathcal{C}}$ and $a_{\mathcal{N}}$ be the meta-actions to select an action from $\mathcal{C}$ and $\mathcal{N}$, respectively, so let $\mathcal{A} = \{a_{\mathcal{C}}, a_{\mathcal{N}}\}$. Therefore there would be a separate softmax layers for actions in $\mathcal{A}$, $\mathcal{C}$, and $\mathcal{N}$ individually. For example, if an agent selects and performs action $a_{\mathcal{C}}$ from $\mathcal{A}$, then the agent consequently selects and performs a classification action $a \in \mathcal{C}$.

Regarding the policy and action selection, each action is selected with probability $\pi(a_t|s_t; \theta)$. Regarding exploration, the agent has $\epsilon$ probability of selecting a random action at each hierarchical level of the action space for some given probability $\epsilon > 0$. This has been incorporated into the algorithm to encourage exploration throughout training.

## 4 IMPLEMENTATION DETAILS

Similar to the work of Deep Q-Networks (Mnih et al., 2013), the raw images from the environment are first preprocessed by converting their RGB representation to gray-scale representation. These converted images are resized to 224 x 224 pixel images. Then the pixels of the images are normalized to floating point values in the range of [0, 1]. Finally, the images are saved in the history. The last $n$ frames of the history are preprocessed and stacked to be produced as input to the Q-function. These are the states in the state space $\mathcal{S}$. The stacked images has a key importance in this problem since state space is dependent on the $n$ number of stacked images.

| Environment output | Condition on action $a_t$ | Possible output value |
|---|---|---|
| state | $a_t \in \mathcal{N}$ | next logical state |
| | else | remain in current state |
| reward | $a_t \in \mathcal{N}$ | 0 |
| | $a_t \in \mathcal{C}$ and correct | 1 |
| | $a_t \in \mathcal{C}$ and incorrect | -1, 0 |
| terminal | $a_t \in \mathcal{C}$ and correct | 0, 1 |
| | $a_t \in \mathcal{C}$ and incorrect | 0, 1 |
| | $t \geq T$ | 1 |

Table 1: Environment output conditioned on action $a_t$ (all the possible environment output after action $a_t$ in state $s_t$)

We created a novel Gym (Brockman et al., 2016) environment from the dataset by simulating an environment with classification and navigation actions with multiple possible configurations. When an agent in state $s_t$ takes an action $a_t$, the environment would read in the next logical image and return the appropriate outputs. The environment outputs a state, reward, and terminal boolean which can be varied depending on the predetermined parameters of the environment, as shown in Table 1. Also, we have a upper bound $T$ on the number of steps taken by the agent per episode, so if $t \geq T$ then the episode is terminated.

Table 1 summarizes all possible environment configurations for the task of joint navigation and recognition. To briefly describe the environment, if the agent performed a movement action i.e. $a_t \in \mathcal{N}$, then the environment would output the next logical state with 0 reward. In case of a classification action i.e. $a_t \in \mathcal{C}$ and it classified correctly, then the environment would output the current state with 1 reward and could either terminate or continue the episode, depending on the environment parameters. Finally, if the action is a classification action i.e. $a_t \in \mathcal{C}$ and it classified incorrectly, then the reward could be either -1 or 0 and could either terminate or continue the episode, depending on the environment parameters. We have tested with multiple environment configurations to determine the best configuration.

The Asynchronous Advantage Actor Critic (A3C) algorithm achieved state-of-the-art results on reinforcement learning task like Atari 2600 games (Mnih et al., 2016). A3C outputs a softmax layer to approximate the policy $\pi$ and a linear layer to approximate the state value $v$. Multiple agents share the weights of the CNN and the agents calculate individual gradients to asynchronously update the weights. Although the CNN instance resides in the CPU for A3C, a GPU-based A3C (GA3C) algorithm (Babaeizadeh et al., 2016) implements the CNN instance in a GPU which showed improvements in computation speed and time over A3C. Compared to training with A3C, GA3C allows for faster training on larger networks and datasets.

Since navigation plays a key role in learning multi-view representations, parameters in the learning algorithm and the environment are set to promote movement. The agent should learn to move efficiently instead of performing frivolous classification actions. We set the discount factor $\gamma = 0.50$ to make the agent learn to classify with lesser number of steps. In the case of a non-hierarchal action space, the n actions have nearly equal probability to occur in the beginning of training. Hence, the agent applies classification for most of the time, since there is a bias towards classification actions. Clearly, the agent shows bias towards classification and results from training this network show minimal movement actions. With the hierarchal action space, the agent has a equal chance of navigation which results in improved movement. We found that hierarchical method gives superior results compared to non-hierarchal action space.

## 5   EXPERIMENTAL EVALUATION

**ModelNet40 Environment:** We convert the rendered Modelnet40 dataset into a Gym environment to evaluate the performance of our method. In each new episode of the environment, the agent would be placed in a randomly selected view of a randomly selected object. The agent can take one of two meta-actions $a_{\mathcal{C}}$ or $a_{\mathcal{N}}$. The meta-actions $a_{\mathcal{C}}$ selects one of 40 actions from $\mathcal{C} = \{a_{airplane}, a_{bathtub}, ..., a_{xbox}\}$ to classify each of the 40 categories and $a_{\mathcal{N}}$ selects one of 2 actions

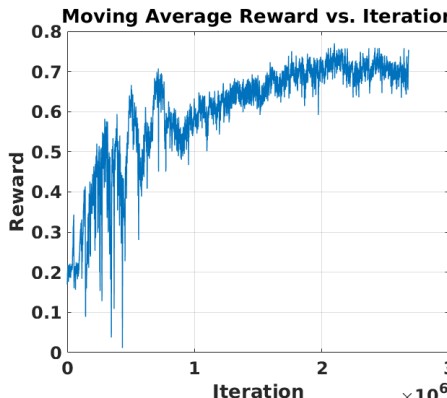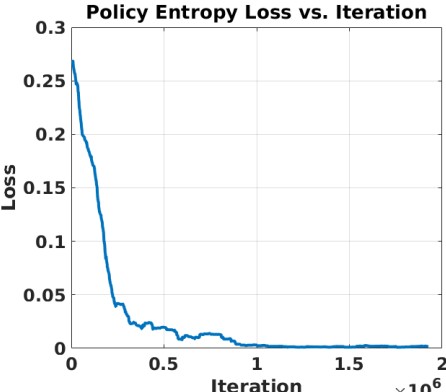

Figure 3: Moving Average Reward vs. Iteration (left) shows the mean reward calculated over a sliding window per iteration. Policy Entropy Loss vs. Iteration (right) shows the negative log of the max policy value per iteration.

from $\mathcal{N} = \{a_{cw}, a_{ccw}\}$ where $a_{cw}$ move around the object in the clockwise direction and $a_{ccw}$ in the counter-clockwise direction. The episode ends when the agent classifies correctly or reaches the maximum number of 12 steps.

We evaluate the method on the ModelNet40 (Wu et al., 2015) dataset as a proof of concept. Model-Net40 has 3D CAD models from 40 categories like airplanes, bathtubs, beds, benches, and more with 3143 training and 760 testing objects. For each object, we create 12 rendered views of by placing 12 cameras around the CAD models every 30 degrees. The cameras are elevated 30 degrees from ground plane and point towards the centroid of the objects.

After training multiple different versions of the ModelNet40 environment with our method, we found the best environment has zero reward for movement and misclassification, one reward for correct classification, and termination after correct classification or max number of 12 steps. Figure 3 shows reward plot of our method and the entropy loss in the policy. The moving average reward is defined as an average reward over a sliding window of 1000 iterations during training. We observe that the reward saturates around 1.5 million iterations. Also, the policy entropy loss is the negative log of the max policy value from the LSD-Net. This decreases over time and converges near zero as shown in the plots. It took nearly 2 million iterations to train the network on a NVIDIA TITAN X GPU.

**THOR and Active Vision Environment:** We use ModelNet40 as a proof of concept and extent the Navigation and detection into a more realistsic environments. The THOR environment consists of multiple rooms created in the virtual environment. While the Active vision dataset was collected from a real world robot exploring the environment with multiple views. The agent can freely navigate the environment simultaneously keeping objects in perspective. To show improvements in the accuracy in object detection in these environments we model the task of object detection as the minimum number of steps taken to recognize the object. This is specifically a challenging task as the agent needs to simulataneouly explore and recognize the object. To train the network faster we give a bounding box around the object to be classified as an input to the network. We observe that our method trains faster compared to previous exploration methods as well as navigate faster to the object compared to the previous state-of-the art results.

### 5.1 QUANTITATIVE EVALUATION

Table 2 shows that our network has learned to classify object with 75.4% average testing accuracy. This was calculated by classifying each group of 12 views for each of the 760 testing objects. The testing accuracy was computed by averaging the accuracy of 20 testing episodes since the starting point for each object is random.

We believe that 3D ShapeNets (Wu et al., 2015) accuracy of 77.3% can be a good baseline to compare our accuracy. Although most of the methods previously proposed show higher accuracy than our method, none of the methods have attempted to solve the retrieval problem just using images. As

| Method | Image(s) | Pose | Depth | Accuracy |
|---|---|---|---|---|
| MVCNN (Su et al., 2015) | ✓ | ✓ | | 90.1% |
| LSD-NET(LSTM) | ✓ | | | **81.4%** |
| DeepPano (Shi et al., 2015) | ✓ | ✓ | ✓ | 77.6% |
| 3D ShapeNets (Wu et al., 2015) | | ✓ | ✓ | 77.3% |
| LSD-NET | ✓ | | | **75.4%** |
| Baseline | ✓ | | | 64.3% |

Table 2: Comparison with other object classification tasks on ModelNet40 with different types of input. Image(s) means the image(s) of the 3D model vs. the full 3D model. Pose is the angle of the object's view. Depth is the depth images of the object.

| Steps Taken | 1 | 2 | 3 | 4 | 5 | 6 | 7 | 8 | 9 | 10 | 11 | 12 | Total |
|---|---|---|---|---|---|---|---|---|---|---|---|---|---|
| Correct class | 118 | 77 | 84 | 47 | 37 | 25 | 19 | 12 | 6 | 4 | 7 | 137 | 573 |
| Wrong class | 76 | 29 | 25 | 11 | 10 | 4 | 2 | 3 | 3 | 2 | 2 | 20 | 187 |
| Correct class(LSTM) | 171 | 90 | 99 | 67 | 59 | 35 | 25 | 17 | 12 | 7 | 8 | 29 | 619 |
| Wrong class(LSTM) | 83 | 16 | 11 | 8 | 5 | 3 | 3 | 1 | 2 | 1 | 2 | 6 | 141 |

Table 3: The number of views encountered i.e. number of steps taken with the number of objects correctly and wrongly classified

another baseline and fair comparison to our method, we trained the same method on the ModelNet environment, but the agent was not allowed to take any movement actions. The baseline method achieved 64.3% accuracy. As seen in Table 2, our method increased in 17.1% accuracy compared to naive classification network.

The agent moved on an average for 619 objects per testing episode and 29 objects took the maximum number of steps. We observe that most of the objects which were classified without movement had rich features to discriminate them like the classes of person, chair, etc. On the contrary, objects which took the maximum steps were symmetric objects and were less distinguishable from multiple views. The classes, like cups and bottles, are view agnostic and took more steps to be classified. But, these view agnostic classes have a overall high classification accuracy as they are very distinguishable from other classes from a single view. Another important conclusion from Table 3 is that the accuracy of classification generally increases with more movement.

We experimented with multiple configurations of the environment. One configuration has negative reward for misclassification to discourage the agent from random classification. Another configuration has positive reward for movement to encourage movement. Both configurations promoted the network to learn only to move instead of classification.

## 5.2 QUALITATIVE EVALUATION

During training, we found that the network first learns to classify the initial images correctly. After it exhausts the classification accuracy from single view, it learns to navigate to improve classification accuracy with multi-view recognition. This is very intuitive and similar to human learning. Humans try to classify the object in the first instance. If the confidence of the object recognition is not high, then they navigate around the object to learn the distinguishable features of the object.

We show some qualitative results of our approach in Figure 4. As seen from the images, the network has learned to move to the best view of the object to accurately recognize it. This is only observed for views which have a bad initialization. We observe that the accuracy improves for objects which have movement compared to objects which were classified without movement. The trend is continuously increasing with the number of steps by the agent. This proves that the network has learned to move to accurately classify and movement is necessary for better classification.

Most methods in Table 2 assume the complete visibility of the object i.e the image rendered from all the views at the same instance from different poses. For example, MVCNN (Qi et al., 2016) passes in all 12 views into a network. In addition, each image has its own convolutional network for feature extraction so MVCNN has 12 convolutional networks. Some methods also use depth information

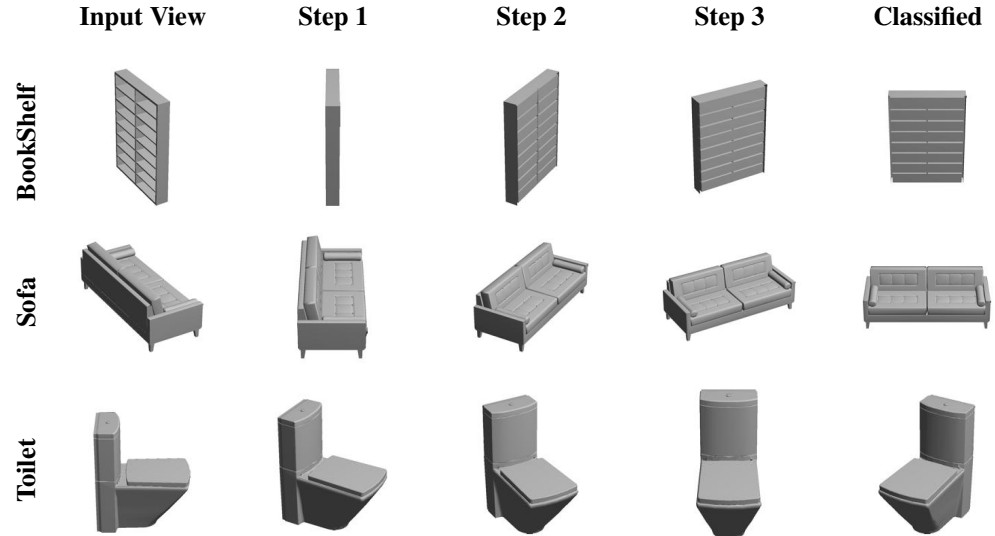

Figure 4: We show a qualitative evaluation of our algorithm on 4 different instances of the environment which took 4 steps before classifying. The first column shows the initial input images and the final image in the sequence classify the object into its respective category.

about the object so the number of views and the number of networks are doubled like in pairwise decomposition (Johns et al., 2016). We believe this comparison is required to show the disadvantages of the other methods. Our method is only dependent on the input images and achieves comparable accuracies to the other methods. Incorporating any of the other cues like pose or depth in a coherent way should boost our accuracy.

In comparison to other methods, our method is agnostic of the starting point i.e. it can start randomly on any image and it would get similar testing accuracies. In a real life setting, it is not necessary for humans to have complete visibility of the object to recognize it. Instead, humans only need to learn as many views as necessary until they are certain. In a similar sense, instead of an inductive bias of which or how many views are passed into the network, the agent learns to navigate to as many views as necessary and classifies the object with good accuracy. Also, our method did not include the use of depth information about the objects, yet resulted in sufficient generalization and accuracy.

Our method performs better better than state-of-the-art in training for navigation to the object. This can be attributed to the fact that object based navigation helps in recognizing the object simultaneously navigate better in the environment. We observe that with oour method the number of steps taken to reach the target has been reduced compared to random walk and object search based exploration.

## 6 CONCLUSION AND FUTURE WORK

We successfully implemented a reinforcement learning based object recognition algorithm that takes classification decisions based on accumulated multi-view information from navigation. We have created a novel environment to simulate the navigation and classification actions. With the LSD-Net framework, we achieved comparative results to state-of-the art accuracies in object recognition with only images as input. The network is currently integrating two tasks, but the architecture can be extended to multi-task learning. We believe that learning in 3D can helping in inferring in 2D images and show some tests on the THOR environment. We currently have not used the geometry of the object from the multiple views during learning. but incorportaing such priors can boost the accuracy of the method.

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
