# OpenReview forum: "LSD-Net: Look, Step and Detect for Joint Navigation and Multi-View Recognition with Deep Reinforcement Learning"
_ICLR.cc/2018/Conference — Reject_

### Official Review · AnonReviewer3 · 2017-11-22
**Interesting problem, but poor evaluation**

**Rating:** 4
**Confidence:** 4

**Review:**

Paper Summary: The paper proposes an approach to perform object classification and changing the viewpoint simultaneously. The idea is that the viewpoint changes until the object is recognized. The results have been reported on ModelNet40.

Paper Strength: The idea of combining active vision with object classification is interesting.

Paper Weaknesses:
I have the following concerns about this paper: (1) The paper performs the experiments on ModelNet40, which is a toy dataset for this task. The background is white and there is only a single object in each image. (2) The simple CNN baselines in MVCNN (Su et al., 2015) achieve higher performance than the proposed model, which is more complicated. (3) The paper seems unfinished. It mentions THOR and Active Vision, but there is no quantitative or qualitative results on them. (4) Some of the implementation details are unclear.

comments:

- It is unfair to use (Ammirato et al., 2017) as the citation for active vision. Active vision has been around for decades.

- It is not clear how the hierarchical soft-max layers have been implemented. There cannot be two consecutive soft-max layers. Also, for example, we cannot select an action from A, and then select an action from C since the operation is not differentiable. This should be clarified in the rebuttal.

- In Table 3, why is there a difference between the performance with and without LSTM in the first column? The LSTM does not see any history at the first step so the performance should be the same in both cases.

- According to Table 1 of MVCNN (Su et al., 2015), a simple CNN with one view as input achieves 83% accuracy (w/o fine-tuning), which is higher than the performance of the proposed method.

- It is better not to call the approach navigation. It is just changing the azimuth of the camera view.

---

### Official Review · AnonReviewer1 · 2017-11-22
**Good paper but deeper evaluation/analysis would make it better.**

**Rating:** 6
**Confidence:** 4

**Review:**


The paper proposes LSD-NET, an active vision method for object classification. In the proposed method, based on a given view of an object, the algorithm can decide to either classify the object or to take a discrete action step which will move the camera in order to acquire a different view of the object. Following this procedure the algorithm iteratively moves around the object until reaching a maximum number of allowed moves or until a object view favorable for classification is reached.

The main contribution of the paper is a hierarchical action space that distinguishes between camera-movement actions and classification actions. At the top-level of the hierarchy, the algorithm decides whether to perform a movement or a classification -type action. At the lower-level, the algorithm either assign a specific class label (for the case of classification actions) or performs a camera movement (for the case of camera-movement actions). This hierarchical action space results in reduced bias towards classification actions.


Strong Points
- The content is clear and easy to follow.
- The proposed method achieves competitive performance w.r.t. existing work.

Weak Points
- Some aspects of the proposed method could have been evaluated better.
- A deeper evaluation/analysis of the proposed method is missing.

Overall the proposed method is sound and the paper has a good flow and is easy to follow. The proposed method achieves competitive results, and up to some extent, shows why it is important to have the proposed hierarchical action space.

My main concerns with this manuscript are the following:

In some of the tables a LSTM variant? of the proposed method is mentioned. However it is never introduced properly in the text. Can you indicate how this LSTM-based method differs from the proposed method?

At the end of Section 5.2 the manuscript states: "In comparison to other methods, our method is agnostic of the starting point i.e. it can start randomly on any image and it would get similar testing accuracies." This suggests that the method has been evaluated over different trials considering different random initializations. However, this is unclear based on the evaluation protocol presented in Section 5. If this is not the case, perhaps this is an experiment that should be conducted.

In Section 3.2 it is mentioned that different from typical deep reinforcement learning methods, the proposed method uses a deeper AlexNet-like network. In this context, it would be useful to drop a comment on the computation costs added in training/testing by this deeper model.

Table 3 shows the number of correctly and wrongly classified objects as a function  of the number of steps taken. Here we can notice that around 50% of the objects are in the step 1 and 12, which as correctly indicated by the manuscript, suggests that movement does not help for those cases. Would it be possible to have more class-specific (or classes grouped into intermediate categories) visualization of the results? This would provide a better insight of what is going on and when exactly actions related to camera movements really help to get better classification performance.
On the presentation side, I would recommend displaying the content of Table 3 in a plot. This may display the trends more clearly. Moreover, I would recommend to visualize the classification accuracy as a function of the step taken by the method. In this regard, a deeper analysis of the effect of the proposed hierarchical action space is a must.

I would encourage the authors to address the concerns raised on my review.

---

### Official Review · AnonReviewer2 · 2017-11-25
**Problem specification unclear; comparison to existing work lacking**

**Rating:** 3
**Confidence:** 4

**Review:**

The ambition of this paper is to address multi-view object recognition and the associated navigation as a unified reinforcement learning problem using a deep CNN to represent the policy.

Multi-view recognition and active viewpoint selection have been studied for more than 30 years, but this paper ignores most of this history.  The discussion of related work as well as the empirical evaluation are limited to very recent methods using neural networks.  I encourage the authors to look e.g. at Paletta and Pinz [1] (who solve a very similar and arguably harder problem in related ways) and at Bowyer & Dyer [2] as well as the references contained in these papers for history and context.  Active vision goes back to Bajcsy, Aloimonos, and Ballard; these should be cited instead of Ammirato et al.  Conversely, the related work cites a handful of papers (e.g. in the context of Atari 2600 games) that are unrelated to this work.

The navigation aspect is limited to fixed-size left or right displacements (at least for ModelNet40 task which is the only one to be evaluated and discussed).  This is strictly weaker than active viewpoint selection.  Adding this to the disregard of prior work, it is (at best) misleading to claim that this is "the first framework to combine learning of navigation and object recognition".

Calling this "multi-task" learning is also misleading.  There is only one ultimate objective (object recognition), while the agent has two types of actions available (moving or terminating with a classification decision).

There are other misleading, vague, or inaccurate statements in the paper, for example:

- "With the introduction of deep learning to reinforcement learning, there has been ... advancements in understanding ... how humans navigate": I don't think such a link exists; if it does, a citation needs to be provided.

- "inductive bias like image pairs": Image pairs do not constitute inductive bias.  Either the term is misused or the wording must be clarified; likewise for other occurrences of "inductive bias".

- "a single softmax layer is biased towards tasks with larger number of actions": I think I understand what this is intended to say, but a "softmax layer" cannot be "biased towards tasks" as there is only one, given, task.

- I do not understand what the stated contribution of "extrapolation of the action space to a higher dimension for multi-task learning" is meant to be.

- "Our method performs better ... than state-of-the-art in training for navigation to the object": The method does not involve "navigation to the object", at least not for the ModelNet40 dataset, the only for which results are given.

It is not clear what objective function the system is intended to optimize.  Since the stated task is object recognition and from Table 2 I was expecting it to be the misclassification rate, but this is clearly not the case, as the system is not set up to minimize it.  What "biases" the system towards classification actions (p. 5)?  Why is it bad if the agent shows "minimal movement actions" as long as the misclassification rate is minimized? No results are given to show whether this is the case or not.  The text then claims that the "hierarchical method gives superior results", but this is not shown either.

Table 3 reveals that the system fails to learn much of interest at all.  Much of the time the agent chooses not to move and performs relatively poorly; taking more steps improves the results; often all 12 views are collected before a classification decision is made.  Two of the most important questions remain open: (1) What would be the misclassification rate if all views are always used? (2) What would be the misclassification rate under a random baseline policy not involving navigation learning (e.g., taking a random number of steps in the same direction)?

Experiments using the THOR dataset are announced but are left underspecified (e.g., the movement actions), but no results or discussion are given.

SUMMARY

Quality: lacking in may ways; see above.

Clarity: Most of the paper is clear enough, but there are confusions and missing information about THOR and problems with phrasing and terminology.  Moreover, there are many grammatical and typographical glitches.

Originality: Harder tasks have been looked at before (using methods other than CNN).  Solving a simpler version using CNN I do not consider original unless there is a compelling pay-off, which this paper does not provide.

Significance: Low.

Pros: The problem would be very interesting and relevant if it was formulated in a more ambitious way (e.g., a more elaborate action space than that used for ModelNet40) with a clear objective function,

Cons: See above.


[1] Lucas Paletta and Axel Pinz, Active object recognition by view integration and reinforcement learning, Robotics and Autonomous Systems 31, 71-86, 2000

[2] Bowyer, K. W. and Dyer, C. R. (1990), Aspect graphs: An introduction and survey of recent results. Int. J. Imaging Syst. Technol., 2: 315–328. doi:10.1002/ima.1850020407

---

### Decision · Program_Chairs · 2018-01-29
**ICLR 2018 Conference Acceptance Decision**

**Decision:**

Reject

**Comment:**

This paper describes active vision for object recognition learned in an RL framework.
Reviewers think the paper is not of sufficient quality: Insufficient detail, and insufficient evaluation.
While the authors have provided a lengthy rebuttal, the shortcomings have not yet been addressed in the paper.